# A Participatory Intervention to Improve the Psychosocial Work Environment and Mental Health in Human Service Organisations. A Mixed Methods Evaluation Study

**DOI:** 10.3390/ijerph18073546

**Published:** 2021-03-29

**Authors:** Emma Cedstrand, Anna Nyberg, Sara Sanchez-Bengtsson, Magnus Alderling, Hanna Augustsson, Theo Bodin, Helle Mölsted Alvesson, Gun Johansson

**Affiliations:** 1Institute for Environmental Medicine, Karolinska Institutet, Unit of Occupational Medicine, 171 77 Stockholm, Sweden; anna.nyberg@pubcare.uu.se (A.N.); magnus.alderling@ki.se (M.A.); theo.bodin@ki.se (T.B.); gun.johansson@ki.se (G.J.); 2Department of Public Health and Caring Sciences, Uppsala University, 751 22 Uppsala, Sweden; 3Center of Occupational and Environmental Medicine, Stockholm Region, 113 65 Stockholm, Sweden; saradelmazo@gmail.com; 4Medical Management Centre, Procome Research Group, Department of Learning, Informatics, Management and Ethics, Karolinska Institutet, 171 77 Stockholm, Sweden; hanna.augustsson@ki.se; 5Department of Global Public Health, Karolinska Institutet, 171 77 Stockholm, Sweden; Helle.Molsted-Alvesson@ki.se

**Keywords:** occupational health intervention, primary organisational intervention, mental health, psychosocial working conditions, process evaluation, effectiveness evaluation

## Abstract

Work-related stress is a global problem causing suffering and economic costs. In Sweden, employees in human service occupations are overrepresented among persons on sick leave due to mental health problems such as stress-related disorders. The psychosocial work environment is one contributing factor for this problem, making it urgent to identify effective methods to decrease stress at the workplace. The aim of the study is to evaluate a participatory intervention to improve the psychosocial work environment and mental health using an embedded mixed methods design. The study is a controlled trial with a parallel process evaluation exploring fidelity and participants’ reactions to the intervention activities, experiences of learning and changes in behaviours and work routines. We collected data through documentation, interviews and three waves of questionnaires. Our results show small changes in behaviours and work routines and no positive effects of the intervention on the psychosocial work environment nor health outcomes. One explanation is end-users’ perceived lack of involvement over the process causing the intervention to be seen as a burden. Another explanation is that the intervention activities were perceived targeting the wrong organisational level. A representative participation over both content and process can be an effective strategy to change psychosocial working conditions and mental health.

## 1. Introduction

Common mental disorders are the predominant cause of sick leave in Sweden today. During the last three decades, the share of psychiatric diagnoses in sick leave spells over two months increased from 13 to 45 percent among women and from 16 to 33 percent among men [1]. Stress-related disorders account for the largest increase among the psychiatric diagnoses, particularly exhaustion syndrome, a diagnosis with a specific ICD-10 code in Sweden, with sick leaves lasting approximately half a year [1]. Further, impaired sleep is a common symptom in common mental disorders, such as depression and exhaustion [2]. In this study, we measure burnout and impaired sleep as symptoms of common mental disorders. Burnout is an undesirable psychological state characterized by exhaustion, cynicism and feelings of reduced professional efficacy [3].

Human service occupations within the health and social care industry typically stand out as risk occupations for sick leave [4]; in particular, sick leave due to stress-related mental disorders [1]. One occupation within these industries with high risk of experiencing high levels of stress contributing to burnout is teachers [5,6].

The reasons for the increase in mental illness are multifaceted, but psychosocial work factors have been found to contribute [7,8]. High job demands, low job control, low social support from co-workers and the supervisor as well as role stress are examples of workplace factors that increase the risk of stress-related disorders. [9]. Cerdas et al. [10] show that there has been a negative development of job demands and decision authority in human service industries such as education, health, and social care in Sweden over time. Within these industries, teachers and elderly care workers are two large occupational groups. High workload and low decision authority have characterized the psychosocial work environment of teachers [11] and home care workers [12] in Sweden. Teachers also report lack of support from management [11] and home care workers experience a sense of being controlled rather than trusted by the management [12]. Interventions to improve the psychosocial working conditions in human service organisations appear warranted.

### 1.1. Primary Interventions to Reduce Work-Related Mental Illness

Organisational interventions targeting stressors in the work environment rather than the individual stress response have been proposed [13,14]. However, even though the amount of research on preventive organisational interventions has increased during the last three decades, it is still relatively scarce according to one systematic meta-review [15] and two systematic reviews [16,17] and findings show inconsistent results [17,18]. Nevertheless, existing research [15,19] on primary interventions suggests that interventions that are comprehensive, also called multi-components programs, tackling many levels at the same time are successful in bringing about positive health effects. Further, preventive interventions targeting enhancement of employee control have shown to have positive effects on mental health [15]. It is argued that the content of complex health interventions should be theory-driven [20,21] to create an understanding of the causal assumptions underpinning the intervention. This is referred to as theory of health, that is, the association between the exposure and the health outcomes [22]. The theory of health is not to be mixed up with the so-called program theory, which describes the association between the intended intervention and changes in exposure/behaviour, in other words, what activities (intervention) are assumed to put the theory of health into practice at the workplace. Further, theories of implementation also need to be considered when designing and evaluating occupational health interventions [19]. These theories could be seen as describing how the activities can be employed in the best way.

A participatory approach has been emphasized in the literature [23,24,25] and means that different stakeholders are involved in, for example, the planning and design of the intervention and/or have a say in the decision of the content of the intervention. However, it is important to clarify in what way participation is applied in intervention studies. Without such clarification, the ability to replicate and draw conclusions about what kind of participatory approach is successful will be limited. Abildgaard et al. [23] have presented a conceptual model of multiple dimensions of a participatory approach, which can be applied when designing or assessing interventions. They suggest a four-dimensional model specifying the aspects of (a) content, (b) process (design, planning, implementation process), (c) directedness, which refers to the use of full participation or representatives and (d) goal. The last dimension reflects whether the participatory approach is a goal in itself or a means to reach a goal, for example, a better work environment.

Organisational interventions are often considered challenging to evaluate. Therefore, it is suggested that elaborative evaluation frameworks are needed in order to uncover the so-called black box [26]. One proposed framework applied in this study is the framework for evaluating organizational-level interventions by Nielsen and Randall [27], which highlights three themes: the intervention and implementation design, the intervention context, and participants’ mental models. The first theme examines to what extent the intervention reached the target group and the latter two represent the factors that may moderate or mediate the link between any intervention exposure and its outcomes. Another process evaluation framework is the Medical Research Council guidance [28], which adopts the same three main dimensions as the Nielsen and Randal framework [27]; however, they also highlight the need of evaluating fidelity, whether or not the intervention was implemented as intended. Further, studies [29,30] have pinpointed the need of not only evaluating changes in working conditions and health outcomes but also the immediate effects such as behaviours and working procedures known as mediators of change. Additionally, whether these immediate achieved changes in turn lead to changes in working conditions and ultimately to changes in health. In Figure 1, the expected order of changes caused by the intervention is presented. First, it is assumed that the participants’ reactions to the intervention activities to some degree need to be positive. The participants also need to perceive that the intervention contributes to increased learning or knowledge. In this case it could be learning about how to systematically work with improvements in the psychosocial work environment. Finally, the participants need to report changes in, for example, behaviours and/or work routines due to the intervention in order to expect changes in distal outcomes such as the psychosocial work environment or health.

### 1.2. Aim and Research Questions

The overall aim of the current study is to evaluate a participatory intervention designed to improve the psychosocial working conditions and decrease symptoms of burnout among teachers and elderly care personnel. Three research questions are included.

Can improvements over time be observed in the psychosocial work environment and health outcomes for the intervention group compared to the control group?Was the intervention implemented as intended (fidelity)?What were the participants’ (a) reactions to the intervention activities, (b) experiences of learning, and (c) changes in behaviours and work routines, related to the intervention (mediators)?

## 2. Materials and Methods

### 2.1. Study Design

We conducted a mixed methods study using an embedded design [31], whereby interviews were carried out alongside three waves of questionnaires. We utilized a controlled trial to evaluate the effectiveness of a primary intervention targeting the psychosocial work environment in two different populations: (a) teachers and (b) elderly care personnel. The design was chosen to handle confounding variables [32]. Further, we applied a parallel process evaluation exploring fidelity and the following mediating factors: (a) reactions to the intervention activities (b) experiences of learning and changes in (c) behaviours and work routines. The Ethical Review Board in Stockholm (registration number 2018/303-31/5) has granted ethical approval.

### 2.2. Study Setting and Participants

The intervention took place in a middle-sized municipality in Sweden from 2016 to 2018. The administrations of early childhood and childhood education and the social services were enrolled. Additionally, the municipality management team and the political boards of each administration were involved. The control groups were matched with workplaces with similar type of work. Within the early childhood and childhood education administration, the elementary school department was chosen to generate both intervention and control groups. The two enrolled schools both represent pre-school to sixth grade. All occupations within the schools were included (intervention: *n* = 60; control *n* = 44) (Figure 2). Within social services, the elderly care department was chosen to generate both intervention and control groups. The home care unit was enrolled as the intervention group (*n* = 82 at baseline) and two nursing home units were invited and accepted to be controls (*n* = 121 at baseline). The intervention and the process evaluation (interviews) include all management levels within the two administrations. The selection for the effectiveness evaluation, however, includes only frontline workers and first-line managers since the aim of the intervention was to improve psychosocial work environment and quality of sleep, and decrease symptoms of burnout in these groups. The activity within the management groups and amongst politicians are seen as means to reach the goal of an improved work environment for frontline workers and first-line managers.

The selection of participants for the interviews was purposefully stratified [33]. The project manager invited the participants. Both those actively engaged in and those more hesitant to the intervention were included. A total of 49 interviews took place. For information about who were interviewed, when they were interviewed and which domains in the Nielsen and Randall framework [27] they were questioned about, see Appendix A. Both principals within the school were asked to participate, but one declined. The two principals were asked to invite two frontline workers from each team and one frontline worker from the administrative team. However, recruitment of teachers was difficult due to teachers’ lack of time, and only one participated. Within the home care department, we invited the unit manager and two (team 1 and 2) out of four team leaders. Two frontline workers from the teams 1 and 2 were also invited for the interviews. Everyone in the home care unit accepted and gave oral consent.

### 2.3. The Intervention

The intervention named “Me, myself, the team and the mission” was designed to promote mental health by enabling a dialogue on workplace challenges associated with stress as identified and prioritized by frontline employees. The job demands–resources theory (JD–R theory) [34] and the effort–reward imbalance model (ERI-model) [35] were used to guide frontline workers and managers in the formation of action plans, encouraging a simultaneous focus on reducing demands, increasing resources and identification of any imbalances regarding efforts and rewards. The intervention was coordinated by Human Resources representatives and one external consultant. The program theory of the evaluated intervention was built on a participatory approach and knowledge development operationalized into five main intervention components. Table 1 shows target groups and aim and content for all intervention components.

The implementation of the intervention focused, in line with research, on engagement from senior [36] and line managers [26,29]. It was initiated and supported by the politicians. The intervention activities among frontline workers were mandatory and held during working hours. When needed, substitutes were summoned.

### 2.4. Data Collection

To evaluate changes in outcomes over time, data were collected using questionnaires at baseline, after 18 and 24 months. Pencil-and-paper surveys were administered in September 2016 (baseline), in February 2017 (follow-up 1) and again in September 2018 (post-intervention). The surveys were distributed by the project manager in the municipality during work meetings. All employees were expected to attend the session, but participation was voluntary. Individuals who were absent on the given time for data collection were given approximately two weeks to fill out the questionnaire. Individual participants were not traceable; however, they were linked to their closest manager, meaning that all analyses are performed using aggregated data.

To evaluate whether the intervention components were delivered according to plan (fidelity) we studied number of intervention activities conducted as planned and proportions attending these activities (dose received). Fidelity was examined using documentation from the project management who kept careful notes of planned and completed activities with completed attendance lists.

Sequential semi-structured individual interviews collected data on reactions, learning and changes in behaviours and in work routines, see Figure 2. Each participant was interviewed up to three times over the duration of the study whereby the first round of the interviews (February 2017) was conducted face to face and phases two (October/November 2017) and three (May 2018) over the telephone. 

The open-ended interview guide was informed by the Nielsen and Randall framework [27] for evaluating organisational-level interventions. The three main domains of the model: contextual factors, intervention and implementation design and mental models were all covered in the interview guide. Changes in routines or behaviours related to the intervention were added since the applied evaluation frameworks lacked this aspect. One chartered psychologist with experience of interviewing conducted all the interviews. Each interview lasted between 10 and 47 min and were recorded by way of an MP3 recorder. The interviews were transcribed verbatim by a transcription service.

### 2.5. Outcome Measures

#### 2.5.1. Primary Outcome

The primary outcome measure is symptoms of burnout, which was determined with the Shirom Melamed Burnout Measurement (SMBM) [37], a 14-item scale from the 21-item long Shirom Melamed Burnout Questionnaire. The measure defines burnout as feelings of physical fatigue (6 items, e.g., “I am physically exhausted”), cognitive weariness (5 items, e.g., “My thinking process is slow”), and emotional exhaustion (3 items, e.g., “I feel like my emotional batteries are dead”). All items were answered on a 7-point rating scale with anchors 1: Almost never and 7: Almost always. Cronbach’s alpha for this scale = 0.95.

#### 2.5.2. Secondary Outcomes and Covariates

Information on psychosocial work environment was collected through 9 scales from the General Nordic Questionnaire (QPSNordic) [38]. For information on how the scales were constructed, which items were included and Cronbach alpha for all scales, see Appendix A. Information on Quality of sleep was collected from one sub-scale of the Karolinska Sleep Questionnaire [39]. Information on educational level, occupation, job tenure, sex, age, part-time/full-time was also included. Additionally, in the second and third waves, we included a question on whether they had answered the previous questionnaire.

### 2.6. Analysis

Differences in background variables, work environment and mental health at baseline between control and intervention groups were tested using independent *t*-tests for continuous variables and Pearson’s chi-square test for categorical variables.

Since the individuals were not traceable, we applied weights and adjusted the analysis for background variables and baseline measures, respectively (cross-sectional analysis). Differences in work environment and mental health at 18 and 24 months between control and intervention groups were tested using independent *t*-tests for continuous variables. However, all outcome data is ordinal, which suggests that the median and inter quartile range should be reported and non-parametric tests be applied. However, normative data [38,40] and previous studies in the field [41,42] report means and standard deviations for the same variables as we apply. Thus, we will do the same to enable comparability between studies. We performed non-parametric tests (Mann–Whitney U-test) correspondingly with the *t*-tests to ensure the trustworthiness of our results. In order to test if homogeneity of variances existed between the intervention and control groups we performed Levene’s test.

Changes of outcome variables over time was calculated using *t*-test for partially overlapping samples in order to handle partially different study population at each time point caused by employee turnover and drop-out [43,44,45]. We also tested for mass significance [46]. See Appendix A for a detailed description of the weighting procedure, comparison between intervention and control groups at different time points, comparison within intervention and control groups at different time points and the way of testing for mass significance. We used SPSS V26 (IBM, Armonk, NY, USA) to analyse quantitative data.

Before starting the analysis, we categorised the imported files in NVivo in order to organise informants by administration to enable identification of different perspectives from these two groups. The two administrations differ in several aspects, for example regarding educational level, which could mean that the intervention was perceived differently within the two administrations. The same was done to identify if the informant was a manager or a frontline worker. Dividing managers and frontline workers occurred because of their different roles in the intervention project. As the frontline workers were the main end-users of the intervention, the aim of the intervention was to improve their work environment. Hence, we compared and contrasted findings from two administrations and two stakeholder groups across time.

The analysis of interview data was organized in two stages. Through both stages, we were guided by Braun and Clarke’s thematic analysis [47], which is a method for identifying, analysing and understanding patterns (themes) within the data. The first stage was guided by the Nilsen and Randall framework [27], using their three broad pre-determined dimensions. The data was initially coded independently by two of the co-authors (EC and SSB) after which they met to discuss using NVivo version 11. To further understand and make sense of the first independent coding, the researchers discussed the coding and revisited the literature in an iterative process. During this process, Kirkpatrick’s learning evaluation model [48] was identified and in a second stage the data was reanalysed with this model in mind. The model includes four levels: (1) reactions to the intervention (2) learning/knowledge (3) behavioural changes and changes in work routines (4) organisational results. However, the fourth level reflects work and health effects, which in this study was assessed by the means of a questionnaire. Hence, the coding in stage two was guided by level one to three of the Kirkpatrick model. The model was chosen because it highlights the linkages between frontline workers and managers reactions to and learning from the intervention activities and the impact of those activities on the outcomes.

After categorising data into the three dimensions of the Kirkpatrick model [48], we searched for themes within these dimensions and reviewed them in relation to the entire data set. After that, the themes were named, defined and related to each other to create an overall story of the data. At last, the research group met to discuss this overall map and final adjustments were made.

## 3. Results

### 3.1. Baseline Participant Characteristics and Outcome Measures

No differences were found in any of the administrations between the intervention and control groups regarding baseline characteristics such as age, gender and job tenure (Table 2). However, the intervention and control groups within both administrations differed significantly concerning several psychosocial working conditions at baseline. Among the schools, the intervention group reported significantly higher social support from superior and control of work pacing at baseline than the control group. For the elderly care, on the other hand, the intervention group reported significantly poorer empowering leadership, social support from superior, role clarity, role conflict, control of work pacing and control of decision (Table 3).

### 3.2. Did the Intervention Improve the Psychosocial Working Conditions, Quality of Sleep and Decrease Symptoms of Burnout?

Regardless of which time points being compared, for both administrations no significant within-group differences over time after adjusting for background variables and the risk of mass significance were found. No data shown.

Regardless of administration, our findings show no significant improvements in the intervention groups compared with the control groups for any of the outcomes, see Table 4. Instead, the results show that all outcome variables within both intervention groups developed in a negative direction compared to the control groups. Further, within the elderly care four secondary outcomes (social support from manager, empowering leadership, control of work pacing and role clarity) differ significantly to the worse for the intervention group at the last follow-up. The primary outcome symptoms of burnout differ significantly to the worse for the intervention group at the last follow-up after adjusting for the baseline measure and the risk of mass significance but not when adjusted for background variables. Further, among the schools, no significant differences at the last follow-up were found. We also performed Mann–Whitney U-tests due to our ordinal data. The results from these tests did not differ from the results from the *t*-tests (Data not shown).

### 3.3. Fidelity

Most of the planned activities targeting frontline workers and first-line managers were delivered according to plan with a high participation rate. However, there is a discrepancy between the management levels, where the senior management and the political level have a lower fidelity compared to the middle and line managers. Due to time constraints and high turnover rates among the senior management, they cancelled most of the planned activities. For a detailed description see Table 5.

### 3.4. Reactions, Learning, Changes in Behaviours and Work Routines

The thematic analysis of the individual interviews led to the development of nine themes with two sub-themes, see Table 6. In general, our findings show both positive and negative reactions to the intervention but hardly any changes in learning, work routines, and or behaviours among frontline workers. However, managers reported changes such as emotional relief and skill development as a result of the coaching component.

Within the home care services managers agreed that the management development had contributed to clarification of goals and roles for the team. Further, frontline workers described increased learning about work routines and where and how decisions are made from participating in the horizontal dialogues. Another encouraging result for this group is the positive reporting of the impact of work groups. This intervention component contributed to both positive emotions and actual changes in behaviours and work routines. Within this activity, the participants perceived a meaningful participation in contrast to the other intervention components.

One theme not depicted in Table 6 represents perceptions about the intervention program as a whole. This theme is named “Not managing expectations” and represents the home care unit only. Both frontline workers and managers explain that they had wished for other types of measures than what was included in the intervention, that is, measures on a structural level such as additional personnel or shorter working hours. Still, the frontline workers report having put quite some hope into the intervention and in its capacity of bringing about better working conditions and a less stressful work environment. Nevertheless, as time passes and no or almost no changes occur, a lot of frustration, mostly among the frontline workers, is expressed.

## 4. Discussion

The results showed, contrary to our expectations, no improvements in psychosocial working conditions or mental health. Instead, the intervention group within elderly care reported a development in an unfavourable direction regarding the psychosocial work environment when compared to the control group. Our results showed that the intervention activities basically were implemented as intended among frontline workers, first-line and middle managers. Low fidelity among these groups can most likely not explain the lack of improvements in outcomes. However, among senior management teams and the political boards, fidelity was low. Furthermore, the intervention led to few changes in learning, behaviours and work routines, which was a prerequisite for improvements in work environment and mental health.

It is often assumed that organisational interventions, although they have no effect, at least do not cause harm [49]. However, other studies [30,50,51] which have evaluated similar organisational interventions as the one evaluated here, also report adverse developments of working conditions or health outcomes in the intervention groups compared to controls. It is important to pay attention to reasons for why some organisational preventive interventions lead to deteriorating working conditions. Below, we will discuss possible reasons for why the present intervention failed to bring about positive change and instead, to some extent, was associated with a negative development.

The studied intervention fulfilled several of the factors identified as important in order to improve psychosocial working conditions and mental health. It had a primary organisational focus, a theoretical starting point regarding theory of health, and a participatory approach. Despite this, it was unsuccessful in bringing about the intended effects. We propose three types of explanations: (1) The design and method of the evaluation did not capture the desired changes (2) Program theory failure (3) Insufficient implementation strategies. These are discussed in more detail below.

### 4.1. Was There a Methodological Failure?

As randomization was not possible, both confounding and selection bias may have affected the reported results. At baseline, the intervention group within elderly care reported less favourable working conditions in five of the nine items, compared to the control groups. It is possible that the intervention group were on a trajectory towards more negative working conditions when the intervention started, a trajectory that the intervention activities were not able to stop. Poorer baseline levels in the intervention group could, on the other hand, be associated with room for improvements, making a positive development more likely [50]. In the schools, the intervention group started out with more favourable scores than their controls on a few outcomes. A possible explanation of the negative development in the schools could, therefore, be attributed to a regression-toward-the-mean effect [52]. Another methodological issue concerns the outcome measures. These were defined and operationalized before the trial. It is possible that, as the intervention project developed and was implemented, the alignment between intervention activities and outcome measures to some extent was lost.

### 4.2. Was There a Program Theory Failure?

The core feature of the program theory was the participatory approach. The stakeholders were involved mainly in developing the content of the intervention. The interviews revealed that the participants perceived the aim of the intervention as meaningful; however, they had less confidence in the form in which it was delivered. The intervention design allowed participation in the development of the content of the intervention, but not over the process. The design, planning and implementation (process) of the intervention was primarily decided by the consultants and HR-representatives resulting in, it seems, a bad organisational fit [21]. Thus, obliging everyone to participate in a fixed process does not seem an effective way to create healthy psychosocial working conditions. It could possibly even be damaging, perhaps explained by the fact that participation increases expectations on the outcome.

Further, it seems that the choice of including all frontline workers instead of using representatives (directedness) was unfortunate. The process evaluation revealed that when representatives were used (vertical dialogues and work groups) the participants reported both positive reactions and actual changes in behaviours and work routines. The analysis also shows that the component (horizontal dialogues) where full participation was applied did not lead to any reported changes in behaviours or work routines, nor was it appreciated (reactions). To the contrary, participants felt that the intervention was very time consuming in relation to what came out of it, which led to frustration and disappointment. Gupta et al. [51] found similar results where the intervention was hypothesized to be seen as a burden instead of an enabler for decreased stress due to the fact that yet another task (the intervention) was introduced.

### 4.3. Was There an Implementation Failure?

The basic question here is whether the intervention activities were carried out as intended (fidelity). Fidelity was high among frontline workers and first-line managers. However, the intervention was not carried out as intended among the senior management teams and the political boards, suggesting that the anticipated support and engagement from them lacked. Another implementation strategy had the aim of enabling everyone to participate by placing the meetings on work hours with summoned substitutes. As most of the intervention components were delivered according to plan, this strategy fulfilled its function. However, it was also associated with high costs for, it seems, little effect.

### 4.4. Deteriorating Trend

Similar to Aust and colleagues [50], the interviews revealed few changes and learning as a result of the intervention activities and a negative trend for all outcome variables within the elderly care. Participants experienced that their expectations were not met by the intervention. They had hoped for more changes in work routines related to the psychosocial work environment and other types of changes, more related to the structural level such as more time to spend with the care recipient (client) and more personnel. It is important to highlight that the working conditions related to structural prerequisites within the home care are challenging to work in [12]. The studied municipality faced problems with high rates of sickness absence and staff turnover when the intervention was initiated [53]. Possibly, it could be damaging to introduce participatory occupational health interventions to workplaces which have undergone great reductions in employee and budgets, thereby lacking the basic prerequisites for a healthy work environment. Mellor et al. [54] refers to this as a lack of organisational capability. One should remember that the intention of the studied intervention was to identify all sorts of psychosocial risks, prioritized by the frontline workers, allowing them to participate and be more involved. However, the obstacles that were related to the structural level (human and technical resources) were never solved (during the two years of the intervention) and it is possible that the decision authority for these obstacles was not even on a municipality level but rather on a national level. Therefore, the hope for improvement was lit, however shaded, which perhaps is psychologically worse than carrying on as usual [50].

### 4.5. Strengths and Limitations of the Study

A strength in the study is the mixed methods design, allowing triangulation of data from the different methods. Without the process evaluation data in this study we would not have been able to understand the results of the controlled trial. Additionally, the longitudinal design with multiple measurements in the controlled trial and the qualitative design (3 phases of interviews) including a large number of informants (interviews) strengthens the study. The controlled trial evaluates the intervention program in total while the qualitative component allows for evaluation of the different intervention components. This gives an opportunity to disentangle the mechanisms of change within the intervention program. One limitation of this study is the lack of identification numbers of the participants, which did not allow to link individuals over time. This means that we cannot investigate within subject changes over time and we cannot control for the possible attrition bias [55]. However, we have tried to compensate for this by considering our overlapping samples, applying the formula by Derrick [44,45] and a weighting procedure, which made it possible to control for confounding variables. 

One further limitation is the selection process of the two interventions groups, allowing possible confounding factors due to differences between intervention and control groups resulting from selection bias [32]. Participants’ perceived changes in routines and behaviours were evaluated through interviews, which implies a risk that the participants adapted their responses to ensure social desirability [56]. To alleviate this risk, the interviewer was not involved in the intervention.

### 4.6. Implications

The participants hoped for structural changes such as more time to spend with the care recipients and more personnel. Thus, they perceived that the intervention was targeting the wrong organisational level. This implies that researchers and practitioners, before implementing an intervention, need to thoroughly explore the context [21] and allow for tailoring of the intervention into the context. Examples of this are the use of formative evaluations with the aim of describing the perceived need for change among the end-users and the use of co-creation as a way to tailor the intervention, take contextual factors into consideration and enhance participation [57,58].

To consider the risk of unintended negative effects of an occupational health intervention, continuously monitoring of these effects are suggested. This is in line with previous research suggesting a dynamic evaluation model for occupational health interventions where results are examined and fed back to the organisation during the trial [59]. If a negative trend is observed, interrupting the intervention should be considered.

## 5. Conclusions

The studied participatory intervention with the aim of improving the psychosocial work environment and mental health failed to bring about the expected changes. We also found some negative trends regarding the psychosocial work environment for the home care unit. We have two suggestions to why. First, even though the program theory was based on a high level of participation the participants influence on the process was perceived as low and frustration over how the intervention was delivered was expressed. This failure contributed to that the intervention was seen as a burden and rendered a bad organisational fit. These consequences contributed to the lack of changes in psychosocial working conditions and mental health. Secondly, our results indicate a lack of fundamental organisational capability in the sense that basic prerequisites such as human and technical resources were insufficient in some respects. Hence, the intervention activities were perceived targeting the wrong organisational level.

Finally, we recommend that the design of future interventions utilize representative participation over both content and process, which may be an effective program theory to change psychosocial working conditions and mental health. This should also be a focus for future studies.

## Figures and Tables

**Figure 1 ijerph-18-03546-f001:**
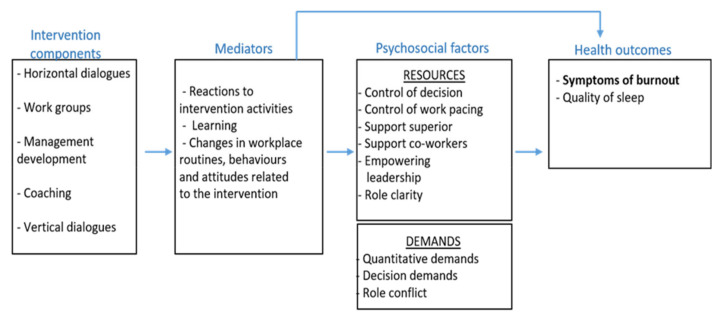
Program logic model of expected order of the changes.

**Figure 2 ijerph-18-03546-f002:**
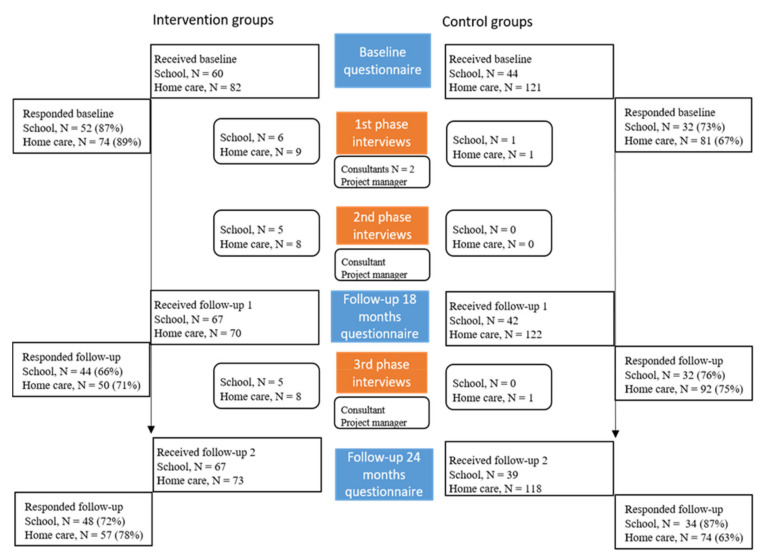
Flow diagram of the workers involved in the effectiveness and process evaluation.

**Table 1 ijerph-18-03546-t001:** Target groups and, aim and content for all intervention components.

Intervention Component	Target Groups	Aim/Content
Kick-off	All employees and politicians	Inform about the project, anchoring.
Joint education	Managers, management teams and politicians	Education on organizational and social work environment.
Feedback and discussion	Municipality management team	Anchoring of the project to enhance support from management.
Feedback and discussion	Political board and the committees representing the two administrations met within each group together with the HR-representative and the external consultant.	Anchoring of the project to enhance support from the politicians. Discussions on the progress of the project and occupational health related issues.
**Leadership coaching**	All team managers (*n* = 5), unit managers (*n* = 3), department managers (*n* = 2) and the two administrative managers.	The overall theme was “leading for health”. However, everyone enrolled in the coaching program formulated his or her own goal for the coaching.
**Horizontal dialogue meetings**	All frontline employees and first-line managers	Make team-specific risk assessments and action plans.
**Management development**	Management teams at 3 levels within the two administrations and the top management team within the municipality.	Clarify goals and create a clear mission for each team. All teams were offered support by the external consultant regarding the topic “leading for health”. A prioritized general focus for these meetings was the clarification of goals and roles, in order to create a clear mission for each team.
**Vertical dialogues**	Chairman of the political committee, managers from all management levels and employee representatives.	Resolve prioritized obstacles within the work environment. Questions discussed for example: parking permissions, subsidized clothes.
**Work groups** (This component was developed during the project. It was not in the original plan)	Representatives from the home care unit.	The work groups involved a couple of frontline workers from the different teams and were led by one team leader together with the external consultant or an HR representative. 5 work groups were organised on the topics: (1) staffing, planning, travel time for the unit (2) our meetings (3) employeeship (4) staffing, planning, travel time for a certain team (5) This is us.
Individual stress management (This component was developed during the project. It was not in the original plan)	All employees at the school.	Individual stress management

The bold names represent the five main intervention components.

**Table 2 ijerph-18-03546-t002:** Characteristics of study participants.

**School**						
	**Baseline**		**18 Months**		**24 Months**	
**Background variables**	**Intervention** ***N* = 52**	**Control** ***N* = 32**	**Intervention** ***N* = 44**	**Control** ***N* = 32**	**Intervention** ***N* = 48**	**Control** ***N* = 34**
Age	*N* (%)	*N* (%)	*N* (%)	*N* (%)	*N* (%)	*N* (%)
<35	10 (21)	7 (22)	11 (25)	9 (28)	9 (19)	10 (29)
36–45	13 (28)	10 (31)	13 (29.5)	12 (38)	17 (35)	12 (35)
>46	24 (51)	15 (47)	20 (45.5)	11 (34)	22 (46)	12 (36)
Education						
Elementary school (9 years)	8 (16)	4 (13.3)	1 (2)	5 (17)	3 (6)	4 (12)
Upper elementary school (>9 years)	5 (10)	2 (6.7)	8 (19)	3 (10)	8 (17)	1 (3)
University/college	38 (74)	24 (80)	34 (79)	22 (73)	37 (77)	29 (85)
Females	36 (70.6)	28 (87.5)	24 (54.5)	26 (87)	32 (67)	28 (82)
Occupation						
Teacher	26 (59)	18 (56.3)	24 (60)	21 (68)	28 (65)	17 (68)
Early childhood educator	4 (9)	4 (12.5)	5 (12.5)	1 (3)	3 (7)	1 (4)
Recreation leader	5 (11)	5 (15.6)	2 (5)	3 (10)	1 (2)	3 (12)
Others	9 (20)	5 (15.6)	9 (28.5)	6 (19)	11 (26)	4 (16)
Work time						
Full time	37 (74)	29 (90.6)	39 (91)	28 (88)	42 (89)	29 (85)
Part time (chosen)	12 (24)	3 (9.4)	3 (7)	3 (9)	4 (9)	5 (15)
Part time (not chosen)	1 (2)	0	1 (2)	1 (3)	1 (2)	0
Job tenure						
<1	16 (31.4)	6 (18.7)	7 (16)	6 (19)	10 (20.8)	4 (12)
1–2	11 (21.6)	4 (12.5)	16 (36)	7 (22)	7 (14.6)	8 (23)
3–5	6 (11.7)	5 (15.6)	3 (7)	3 (9)	10 (20.8)	7 (21)
>5	18 (35.3)	17 (53.2)	18 (41)	16 (50)	21 (43.8)	15 (44)
Answered before (at 24 months)					*N* (%)	*N* (%)
Yes, both times					24 (50)	16 (47.1)
Answered one of them					8 (17)	11 (32.4)
No, none of them					11 (23)	6 (17.6)
Do not remember					4 (8)	1 (2.9)
Missing					1 (2)	0
**Elderly Care**						
	**Baseline**		**18 Months**		**24 Months**	
**Background variables**	**Intervention** ***N* = 74**	**Control** ***N* = 81**	**Intervention** ***N* = 50**	**Control** ***N* = 92**	**Intervention** ***N* = 57**	**Control** ***N* = 74**
Age	*N* (%)	*N* (%)	*N* (%)	*N* (%)	*N* (%)	*N* (%)
<35	16 (25)	12 (21)	11 (22)	11 (12)	15 (26)	7 (10)
36–45	13 (20)	5 (8)	11 (22)	19 (21)	13 (23)	15 (20)
>46	36 (55)	41 (71)	27 (56)	70 (67)	29 (51)	51 (70)
Education						
Elementary school	9 (12.7)	10 (14)	6 (12)	18 (20)	6 (10.5)	14 (19)
Upper elementary school	51 (71.8)	53 (75)	33 (69)	61 (68)	39 (68.5)	49 (66)
University/college	11 (15.5)	8 (11)	9 (19)	11 (12)	12 (21)	11 (15)
Females	66 (90)	73 (94.8)	42 (89)	80(89)	49 (86)	62 (84)
Occupation						
Care assistant	23 (34.8)	16 (22)	11 (33)	12 (16.5)	19 (41)	11 (19)
Assistant nurse	43 (65.2)	56 (78)	21 (64)	60 (82)	27 (59)	48 (81)
Nurse	0	0	1 (3)	0	0	0
Cleaner	0	0	0	1 (1.5)	0	0
Work time						
Full time	42 (58)	39 (53.5)	22 (46)	49 (56.5)	28 (51)	37 (53)
Part time (chosen)	28 (39)	31 (42.5)	26 (54)	35 (40)	27 (49)	32 (45.5)
Part time (not chosen)	2 (3)	3 (4)	0	3 (3.5)	0	1 (1.5)
Job tenure						
<1	10 (13.7)	10 (13)	4 (8)	7 (8)	4 (7)	5 (7)
1–2	9 (12.3)	13 (17)	3 (6)	14 (15)	3 (5)	12 (16)
3–5	15 (20.5)	8 (10)	8 (16.5)	12 (13)	16 (28)	11 (15)
>5	39 (53.5)	46 (60)	34 (69.5)	58 (64)	34 (60)	46 (62)
Answered before (at 24 months)					*N* (%)	*N* (%)
Yes, both					34 (60)	31 (42)
Answered one of them					4 (7)	16 (22)
No, none of them					4 (7)	7 (10)
Do not remember					10 (17.5)	16 (22)
Missing					5 (8.5)	4 (4)

**Table 3 ijerph-18-03546-t003:** Mean and Standard deviation for all outcome variables at all three time points.

**School**						
	**Baseline**		**18 Months**		**24 Months**	
	**Intervention** ***N* = 52**	**Control** ***N* = 32**	**Intervention** ***N* = 44**	**Control** ***N* = 32**	**Intervention** ***N* = 48**	**Control** ***N* = 34**
	**M (Sd)**	**M (Sd)**	**M (Sd)**	**M (Sd)**	**M (Sd)**	**M (Sd)**
Primary outcome						
Burnout (1–7)	2.5 (1.2)	2.1 (1.1)	2.5 (1.2)	2.3 (1.1)	2.7 (1.2)	2.1 (1.0)
Secondary outcomes						
Role conflict (1–5)	2.5 (0.8)	2.5 (0.7)	2.7 (0.8)	2.3 (0.8)	3.0 (0.8)	2.6 (0.6)
Role clarity (1–5)	4.4 (0.6)	4.5 (0.5)	4.2 (0.9)	4.7 (0.5)	4.1 (0.8)	4.6 (0.5)
Social support manager (1–5)	4.4 * (0.6)	4.0 (0.7)	3.7 (1.0)	4.2 (0.9)	3.4 (1.0)	4.1 (0.6)
Empowering leadership (1–5)	3.6 (1.1)	3.3 (0.7)	3.0 (1.0)	3.5 (0.9)	2.9 (1.1)	3.4 (0.7)
Social support colleagues (1–5)	4.5 (0.6)	4.3 (0.7)	4.2 (0.6)	4.3 (0.7)	4.0 (0.7)	4.5 (0.6)
Control of decision (1–5)	2.9 (0.6)	2.9 (0.6)	2.9 (0.5)	2.8 (0.6)	2.7 (0.6)	2.8 (0.6)
Control of work pacing (1–5)	2.3 * (0.8)	1.8 (0.5)	2.4 (0.9)	1.8 (0.6)	2.2 (0.9)	2.0 (0.7)
Quantitative job demands (1–5)	2.9 (1.0)	3.0 (0.9)	3.0 (0.8)	3.0 (0.8)	3.3 (1.0)	3.1 (0.7)
Decision demands (1–5)	3.9 (0.7)	4.0 (0.6)	3.9 (0.6)	4.0 (0.6)	4.0 (0.6)	4.0 (0.5)
Quality of sleep (1–6) ^1^	2.8 (0.8)	2.6 (1.1)	2.8 (1.0)	2.8 (0.8)	2.9 (0.9)	2.7 (0.8)
**Elderly Care**						
	**Baseline**		**18 Months**		**24 Months**	
	**Intervention** ***N* = 74**	**Control** ***N* = 81**	**Intervention** ***N* = 50**	**Control** ***N* = 92**	**Intervention** ***N* = 57**	**Control** ***N* = 74**
Primary outcome						
Burnout (1–7)	2.4 (1.1)	2.2 (1.2)	2.4 (1.0)	1.9 (1.1)	2.9 (1.2)	1.9 (0.9)
Secondary outcomes						
Role conflict (1–5)	2.9 * (0.7)	2.6 (0.9)	3.0 (0.6)	2.4 (0.9)	3.0 (0.8)	2.3 (0.9)
Role clarity (1–5)	3.9 * (0.8)	4.5 (0.5)	4.0 (0.8)	4.6 (0.5)	4.1 (0.6)	4.6 (0.6)
Social support manager (1–5)	3.8 * (0.8)	4.2 (0.9)	3.5 (0.9)	4.3 (0.7)	3.4 (1.0)	4.3 (0.8)
Empowering leadership (1–5)	3.0 * (1.1)	3.6 (1.0)	2.5 (0.9)	4.0 (1.0)	2.6 (1.1)	3.9 (1.0)
Social support colleagues (1–5)	4.2 (0.8)	4.3 (0.7)	4.2 (0.8)	4.3 (0.7)	4.2 (0.7)	4.2 (0.9)
Control of decision (1–5)	2.3 * (0.5)	2.7 (0.8)	2.3 (0.6)	2.9 (0.8)	2.3 (0.7)	2.9 (0.8)
Control of work pacing (1–5)	2.1 * (0.6)	2.5 (0.8)	2.0 (0.8)	3.0 (0.8)	2.0 (0.8)	2.9 (0.8)
Quantitative job demands (1–5)	2.9 (0.7)	2.7 (0.8)	3.0 (0.7)	2.5 (0.9)	3.0 (0.7)	2.5 (0.8)
Decision demands (1–5)	3.6 (0.7)	3.7 (0.8)	3.5 (0.7)	3.3 (0.8)	3.6 (0.6)	3.3 (0.7)
Quality of sleep (1–6) ^1^	2.7 (0.7)	2.7 (1.1)	2.7 (0.8)	2.6 (1.0)	3.1 (0.8)	2.5 (1.0)

^1^ Higher score indicate lower quality of sleep. * *p* < 0.05.

**Table 4 ijerph-18-03546-t004:** Between group differences at second and third follow-up.

**School**								
	**Experiment vs. Control 18 Months**				**Experiment vs. Control 24 Months**			
	**Weighted for** **Background Variables**		**Weighted for** **Baseline Measure**		**Weighted for** **Background Variables**		**Weighted for** **Baseline Measure**	
Primary outcome	Mean diff.	*p*-value ^1^	Mean diff.	*p*-value ^1^	Mean diff.	*p*-value ^1^	Mean diff.	*p*-value ^1^
Burnout	0.39	1.0	0.12	1	0.95	0.13	0.31	1.0
Secondary outcomes								
Role conflict	−0.53	0.13 *	−0.13	1.0	−0.93	0.001	−0.24	1.0
Role clarity	−0.17	1.0	−0.24	1.0	−0.62	0.001	−0.27	0.66
Social support manager	−0.91	0.001 *	−0.25	1.0	−1.18	0.001	−0.28	1.0
Empowering leadership	−0.85	0.001	−0.35	1.0	−0.92	0.001	−0.06	1.0
Social support colleagues	−0.24	1.0	−0.12	1.0	−0.59	0.001	−0.25	1.0
Control of decision	−0.31	0.66	0.06	1.0	−0.33	1.0 *	−0.01	1.0
Control of work pacing	0.61	0.26	0.22	1.0	−0.18	1.0 *	0.02	1.0
Quantitative job demands	0.29	1.0	0.06	1.0	0.62	0.53	0.12	1.0
Decision demands	0.60	0.001 *	0.04	1.0	0.14	1.0 *	0.02	1.0
Quality of sleep	0.16	1.0 *	−0.04	1.0	0.51	1.0	−0.06	1.0
**Elderly Care**								
	**Experiment vs. Control 18 Months**				**Experiment vs. Control 24 Months**			
	**Weighted for** **Background Variables**		**Weighted for** **Baseline Measure**		**Weighted for** **Background Variables**		**Weighted for** **Baseline Measure**	
Primary outcome	Mean diff.	*p*-value ^1^	Mean diff.	*p*-value ^1^	Mean diff.	*p*-value ^1^	Mean diff.	*p*-value ^1^
Burnout	0.40	1.0	0.37	1.0	0.75	0.13	0.62	0.001 *
Secondary outcomes								
Role conflict	−0.67	0.001	−0.24	1.0 *	−0.67	0.001	−0.35	0.13
Role clarity	−0.41	0.13 *	−0.38	0.001 *	−0.45	**0.001**	−0.37	**0.001**
Social support manager	−1.03	0.001	−0.66	0.001	−0.78	**0.001**	−0.65	**0.001**
Empowering leadership	−1.29	0.001 *	−1.03	0.001	−1.08	**0.001**	−0.80	**0.001**
Social support colleagues	0.26	1.0	−0.02	1.0	0.29	1.0 *	−0.15	1.0 *
Control of decision	−0.30	0.40	−0.26	0.66	−0.35	0.13	−0.30	0.13
Control of work pacing	−0.67	0.001	−0.51	0.001	−0.79	**0.001**	−0.50	**0.001**
Quantitative job demands	0.54	0.001	0.42	0.001	0.32	1.0 *	0.36	0.001
Decision demands	0.65	0.001	0.12	1.0	0.12	1.0	0.22	1.0
Quality of sleep	−0.02	1.0	0.04	1.0	0.29	1.0	0.31	1.0

^1^ All *p*-values are Bonferroni-adjusted. * Indicates different variances between the intervention and control groups as a result of the Levene’s test *p* < 0.05. Outcomes significant after adjusting for background variables and the baseline measure at 24 months are in bold.

**Table 5 ijerph-18-03546-t005:** Dose delivered (numbers planned/delivered) and dose received (N/%) for all intervention components.

**Type of Intervention Component**	**No. Planned/Delivered**	**Dose Received** ***N*/%**
Kick-off (all employees and politicians)	2 times × 2 h/2 times × 2 h	149/not available
Joint education (managers, management teams and politicians)	6 times, 2 parts. Each part was supposed to be delivered 3 times each/Part one was delivered twice and part two was delivered once.	Part one = 11/not available
Feedback and discussion (municipality management team)	7 times × 1 h/2 times × 75 min	Not available
Feedback and discussion (political board)	7 times × 1 h/2 times × 2 h	Not available
**Leadership coaching**	School 6 h per person/4.3 h per person Social services 6 h per person/5.4 h per person	Not available
**Horizontal dialogue meetings (workshops in the team or at the unit)**	School 7 workshops per team/7 workshops per team Social services 7 workshops per team/6–8 among the four teams	Mandatory
**Management development (management teams at 3 levels)**	School Senior 12 h/2 times × 2 h Middle 12 h/3 times × 2 h First-line 12 h/15 times × 2 h Social services Senior 12 h/4 times × 2 h Middle 12 h/5 times × 3 h First-line 12 h/26 times × 2 h	Not available
**Vertical dialogues** **(with politicians, managers and employee representatives)**	School Not planned/7 times × 1 h Social services Not planned/5 times × 1 h	
**Work groups**	Not planned 5 groups, 28 times ×2 h (4–8 meetings per group).	
Individual stress management	Not planned/8 times × 90 min	

The bold names represent the five main intervention components.

**Table 6 ijerph-18-03546-t006:** Main findings from the thematic analysis.

Type of Intervention Component	Reactions	Learning	Changes in Behaviours and Routines
Leadership coaching	**Themes:** Rewarding (emotional relief) Appreciated, meaningful. They report getting support in resolving complicated situations with employees, co-workers or their closest manager.	**Themes:** Rewarding (skill development) Contributed to both personal and professional development. Improved skills in conflict management, how to organize meetings and how to coach employees are reported.	
Horizontal dialogue meetings (workshops in the team or at the unit)	**Themes:** Lack of change and disagreement on the format. Disappointment due to the lack of changes coming out from the project. Positive about the possibility to listen to each other and discuss work-related challenges (all). Frustration about how the meetings are organised.	**Themes:** Better understanding of work routines and colleagues. How decisions formally are taken and who is responsible for what (home care).	
Management development (management teams at 3 levels)		**Themes:** Mutual mental models (home care) Helped them clarify the goals and roles of the team, giving its members a mutual understanding of each other and of what they wish to achieve	**Themes:** Mutual mental models (home care) The four work groups operate more equal, applying the same work routines more often than before
Vertical dialogues (with politicians, managers and employee representatives)	**Themes:** A sense of being listened to Appreciated the opportunity to communicate the needs of the organisation directly to the politicians.		**Themes:** Potentially moving towards change One example of an actual change in work routines due to the vertical dialogues is reported. It is within the home care where the suggestion of purchasing winter jackets for the workers has been approved.
Work groups	**Themes:** Meaningful participation Experiences of being listened to and taken seriously (involved).		**Themes:** Meaningful participation Introduction of dedicated time to write and read reports from a day’s work.

## Data Availability

The data presented in this study are available on request from the corresponding author. The data are not publicly available due to legal restrictions.

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
