# Peer review of "A Participatory Intervention to Improve the Psychosocial Work Environment and Mental Health in Human Service Organisations. A Mixed Methods Evaluation Study"

_ijerph, 2021, doi:10.3390/ijerph18073546_

Round 1

Reviewer 1 Report

The research is very interesting. The research should be repeated in a few years to compare the results. 

Author Response

We did not receive any suggested revisions from this reviewer, hence no response could be given.

Reviewer 2 Report

The study investigates the effectiveness of multi-level interventions in two different occupational sectors through a mixed-method approach. Even though the participatory intervention failed to bring the expected changes, the process evaluation analysis highlights critical components which might explain the unexpected results.

In general, the study is exciting and methodologically straightforward. However, there are main concerns and open points that must be addressed to improve the paper's quality. The reading is tough to follow, and the use of the tables is detrimental to support the reader. The authors are encouraged to follow the suggestions provided below. 

The introduction section touches on several aspects, but none seems to be adequately defined and explained, which creates confusion. It might be not sufficient to mention some systematic reviews without introducing your dimensions properly. It appears that burnout and the quality of sleep are the study's primary health outcomes, but very few aspects have been presented about those dimensions. Moreover, as a secondary outcome, social support from the manager, empowering leadership, control of work pacing and role clarity are not properly introduced in the paper.

The authors should describe the dimensions involved in the process evaluation more in detail. It is not clear why the authors describe some dimensions while others not. The same goes for the proposed framework evaluating the intervention are cited but not adequately defined. 

The paper discusses the challenge to evaluate an organisational intervention, whereas, in the study, it seems the authors adopt a multi-level approach. 

Figure 1 is introduced without a proper description of the causal relationship and the boxes' examples. It might be beneficial to provide concrete examples in the text with the concepts represented in the figure. 

The 2.3 section is very complicated to understand readily. See the specific comment below for suggestions.

Again, Braun and Clarke's thematic analysis and Kirkpatrick's model are just mentioned to explain the completion of stage two, but a thorough description of this stage might be beneficial.

Tables are difficult to read. Please try to elaborate on different ones looking for consistency. For example, in table 1, the measurement time point is in columns, while in table 2, it is in rows, making it very complicated to read the values for each dimension.

Is there a reason why in table 3, some p-values are in bold?

Table 4 is tough to follow and read. First, it might be beneficial to present a table with the different type of intervention, the target involved, and each implementation phase's aims in the devoted section (2.3 The intervention). Then, it might be helpful to divide table 4 into two, the first one with objective data on dose planned, dose delivered, and dose received, and the second one with the qualitative key findings. 

Minor issues:

The introductory part of section 1.1 should make it explicitly clear that the few studies mentioned refer to systematic reviews or meta-analyses. Stating that few studies have evaluated organisational interventions may seem misleading.

Line 359 might need a reference for the regression-toward-the-mean effect.

Author Response

We are grateful to your relevant and useful comments. We carefully considered and responded to these comments as you will find in the attached file.

Reviewer 3 Report

Introduction

-> Describe the reason why it is needed to study stress collectively rather than individually in primary interventions to reduce work-related mental illness of this study, based on the preceding studies.

-> Add the previous studies, which served as criteria for selection of teachers and elderly care personnel that this study targeted.

-> There are problems with research questions of this study. Check the below and revise.

1) Previous studies have already stated that reinforced control-based prevention positively influenced mental health. Why did study choose a control group?

2) There should be a deeper explanation of the intervention in the text.

Materials and Methods

-> Present a definite survey and interview date of each group in this study.

-> Further information about the questionnaire and interview is required.

-> Add skewness and kurtosis values of the factors used in this study. This is to secure validity of t-test conducted for parametric statistics.

Results

-> After a t-test, results of homogeneity of variance test are necessary to verify normality of relevant samples. Thus, present the results of homogeneity of variance test following all t-tests.

-> There should be a detailed explanation of why and how individual interview answers were subdivided.

Discussion and Conclusion

-> According to the discussion, it cannot help in improving psychosocial working conditions and mental health. There should be a full explanation of how this result came out.

-> It says these study results are different from expectations. Thus, present implications of these study results.

-> The sentence ‘Was there an implementation failure?’ shows the study results differ from expectations due to insufficient research participation. I consider this is because the researcher did not fix proper criteria when conducting the survey and interview. Therefore, there is a need to describe limitations and problems of this study clearly.

Author Response

(The authors gave the same response as above.)

Round 2

Reviewer 2 Report

I would like to thank the authors for responding adequately to all my comments. The quality of the article has improved considerably and in its current version it is accepted for publication.